# Plasma Neurofilament Light Chains as Blood-Based Biomarkers for Early Diagnosis of Canine Cognitive Dysfunction Syndrome

**DOI:** 10.3390/ijms241813771

**Published:** 2023-09-06

**Authors:** Chung-Hsin Wu, Xun-Sheng Pan, Li-Yu Su, Shieh-Yueh Yang

**Affiliations:** 1School of Life Science, National Taiwan Normal University, Taipei 106, Taiwan; 2Zaixing Animal Hospital, Taipei 116, Taiwan; michaelvet56@yahoo.com.tw; 3Department of Physiology, College of Medicine, National Taiwan University, Taipei 106, Taiwan; julia10025@gmail.com; 4MagQu Co., Ltd., New Taipei City 112, Taiwan; syyang@magqu.com

**Keywords:** canine, cognitive dysfunction, neurofilament light chain, immunomagnetic reduction

## Abstract

The number of elderly dogs is increasing significantly worldwide, and many elderly dogs develop canine cognitive dysfunction syndrome (CCDS). CCDS is the canine analog of Alzheimer’s disease (AD) in humans. It is very important to develop techniques for detecting CDDS in dogs. Thus, we used the detection of neurofilament light chains (NfL) in plasma as a blood-based biomarker for the early diagnosis of canine Alzheimer’s disease using immunomagnetic reduction (IMR) technology by immobilizing NfL antibodies on magnetic nanoparticles. According to the 50-point CCDS rating scale, we divided 36 dogs into 15 with CCDS and 21 without the disease. The results of our IMR assay showed that the plasma NfL levels of dogs with CCDS were significantly increased compared to normal dogs (*p* < 0.01). By plasma biochemical analysis, we further confirmed that the liver and renal dysfunction biomarkers of dogs with CCDS were significantly elevated compared to normal dogs (*p* < 0.01–0.05). On the basis of our preliminary study, we propose that IMR technology could be an ideal biosensor for detecting plasma NfL for the early diagnosis of CCDS.

## 1. Introduction

Under the influence of factors such as the global birth rate and the extension of human life expectancy, there has been a wave of pet keeping, raising the problem of the long-term care of and medical expenses for aging pets. The population of older dogs is currently growing significantly, and canine cognitive dysfunction syndrome (CCDS) is a progressive neurodegenerative disorder in older dogs that is often attributed to the natural process of aging [1]. Older dogs are at higher risk of developing CCDS, but there is currently no effective treatment. Elderly dogs with CCDS exhibit cognitive deficits typical of Alzheimer’s disease (AD) in humans [2]. Dog owners are often insufficiently aware of behavioral changes in their dogs, and ignore the early signs of CCDS. It is often too late when elderly dogs are diagnosed with CCDS. In elderly dogs with CCDS, the number of neurons in the cerebral cortex, hippocampus, and limbic system is significantly reduced [3,4]. CCDS is mainly seen in older dogs over the age of 10 years, which usually show slow behavioral and cognitive changes. The CCDS rating scale (CCDR) can be used to discern cognitive decline in elderly dogs [3].

Currently, the essential diagnostic tools for CCDS mainly rely on screening biomarkers in blood and cerebrospinal fluid (CSF) [4,5]. Because most owners are reluctant to have CSF extracted from their dogs, blood samples remain an appropriate method for testing CCDS biomarkers [6,7]. Unlike blood tests for AD in humans, the changes in plasma Aβ_1-42_ level and amyloid deposits in brain tissue in older dogs with CCDS are not obvious [8,9]. It is necessary to find other candidate biomarkers, such as neurofilament light protein (NfL). On the basis of clinical research into AD, it has been found that plasma NfL levels can be used as biomarkers for monitoring neurodegeneration and disease progression in the pre-symptomatic stage of AD [10]. In veterinary testing, few studies have been conducted on biomarkers associated with CCDS. Biomarkers such as plasma NfL that are suitable for routine clinical testing must be readily detectable. The NfL concentration can be easily measured in plasma; therefore, it is a potential biomarker for veterinary testing in brain tissue damage [11].

The detection of NfL in CSF has been proven to be a useful tool in diagnosing various types of neurodegenerative diseases, such as early onset AD [12]. However, collecting CSF requires lumbar puncture and can cause many uncomfortable side effects; furthermore, it is not easy to clinically detect NfL in the CSF of dogs. Fortunately, there is another possibility for identifying CCDS, through the detection of NfL in blood. However, the concentration of NfL in blood is extremely low, rendering it almost undetectable using traditional assays. With the development of ultrasensitive detection techniques, the accurate detection of NfL in dog plasma has become possible [13,14]. A recently published paper showed that immunomagnetic reduction (IMR) could be used to detect NfL concentrations in plasma. The NfL antibody was immobilized on magnetic nanoparticles, and then NfL concentrations were determined in the plasma of normal controls and patients with Parkinson’s disease (PD) or AD. It was found that there were significant differences in plasma NfL concentrations among the normal controls, PD patients, and AD patients [15]. These published results demonstrate the potential clinical impact of using plasma NfL to diagnose AD or PD. Our study aims to explore the preclinical performance of IMR in combination with NfL reagents for the early diagnosis of CCDS.

## 2. Results

### 2.1. Canine CCDR Scores

To clearly distinguish between physiological and pathological aging in dogs, we used the CCDR, which was developed by Salvin et al. [5]. As shown in Figure 1, 36 small- and medium-sized dogs were divided, according to the CCDS rating scale of 50 points, into 15 dogs with CCDS (CCD) and 21 without the disease (ND). Dogs were indicated as having CCD if they had a CCDR score ≥50. The CCDR scale has a high diagnostic accuracy (99.3%) for the detection of dogs with CCDS [5]. We observed behavioral changes in the 15 dogs with CCDS, which were mostly related to disruption and irritability in the relationship between the owners and the dogs. In addition, those dogs with CCDS also showed abnormal responses to familiar objects, aimless wandering, and abnormal reactions when recognizing familiar people and animals.

### 2.2. Canine Plasma NfL Levels 

We used a highly sensitive 36-channel high-Tc SQUID alternating current (ac) magnetosusceptometer that is able to detect ultra-low concentrations of NfL in plasma (Figure 1, right). Then, we established a curve of the relationship between the IMR signals and plasma NfL expression, as shown in Figure 2. We found that the plasma NfL expression was positively correlated with IMR signal (as percentage) at concentrations between 0.1 and 10,000 pg/mL, following the logistic function. The data point at 10,000 pg/mL NfL still lies on the fitted line. This implies that the hook effect does not occur at 10,000 pg/mL NfL. The hook effect refers to a surplus of antibodies or antigens. When the concentration of antibodies or antigens is very high, the ability of the antibodies to form immune complexes is compromised [15].

According to the relationship curve in Figure 2, we quantified and compared the plasma NfL levels of 36 dogs in Table 1. According to the CCDR of 50 points, we divided 36 dogs into 21 ND and 15 into CCDS. Our data showed that the average age of the ND group was 7.32 years for males and 7.68 years for females, while the average age of the CCD group was 14.37 years for males and 13.87 years for females (ND vs. CCD, *p* < 0.01). Furthermore, the average plasma NfL level of the ND group was 7.79 pg/mL for males and 8.02 pg/mL for females, while the average plasma NfL level of the CCD group was 9.37 pg/mL for males and 9.92 pg/mL for females (ND vs. CCD, *p* < 0.01).

### 2.3. Canine Blood Test 

As shown in Table 2, we used IDEXX kits to examine canine complete blood count and serum biochemical parameters. Our data showed that canine complete blood counts (such as hematocrit, hemoglobin, white blood cell, eosinophil, granule, and platelet counts) had no significant differences between the ND and CCD groups. As for canine serum biochemical parameters, the quantified plasma renal dysfunction biomarkers (such as symmetrical dimethylarginine, creatinine, urea nitrogen, total protein, albumins), hepatocellular necrosis biomarkers (such as aspartate transaminase (AST), alanine transaminase (ALT), cholestasis, alkaline phosphatase (ALP), glutamyl transferase (GMT)), and inflammation biomarkers (such as C-reactive protein) of the CCD group were significantly increased compared with the ND group (CCD vs. ND, *p* < 0.01–0.05). 

## 3. Discussion

In this study, we mainly used the IMR assay platform to detect the concentration of plasma NfL in dogs. According to the CCDS rating scale of 50 points, we divided 36 dogs into 15 with CCDS and 21 without CCDS. By a highly sensitive 36-channel high-Tc SQUID alternating current (ac) magnetosusceptometer, our results showed that the plasma NfL levels of dogs with CCDS were significantly increased compared to those without CCDS (*p* < 0.01). Using plasma biochemical analysis, we further confirmed that liver and renal dysfunction biomarkers of dogs with CCDS were significantly elevated compared to those without CCDS (*p* < 0.01–0.05). From our preliminary study, we suggest that IMR technology should be an ideal biosensor in detecting plasma NfL for the early diagnosis of CCDS.

According to our previous study [15], using NfL detection reagents on the IMR analysis platform can show ultra-sensitive detection results at the fg/mL level. Table 3 shows the detection limits of the NfL concentration in various assay platforms, such as enzyme-linked immunosorbent assay (ELISA), single-molecule assay (SIMOA), electrochemiluminescence (ECL), and IMR. The detection limits of NfL concentration of ELISA, SIMOA, and ECL only show a level of sensitivity at pg/mL. ECL is more sensitive than ELISA for detecting the NfL concentration but not as sensitive as SIMOA. The detection limits of the NfL concentration using IMR assays are at least 1000-fold more sensitive than ELISA, SIMOA, or ECL. Notably, the use of IMR to detect NfL concentrations in canine plasma is not only ultrasensitive but also specific.

Using IMR assay, our results showed that the plasma NfL levels of dogs with CCDS significantly increased compared to those without CCDS (8.01 vs. 9.66 pg/mL, *p* < 0.01). The cut-off values of plasma NfL levels were 8.84 pg/mL for differentiating between dogs with and without CCDS. As suggested in our study, those dogs with NfL levels ≥8.84 pg/mL should be at risk for CCDS. However, our experimental data also showed that some dogs with NfL levels ≥8.84 pg/mL did not have symptoms related to CCDS, and some dogs with NfL levels <8.84 pg/mL did have symptoms related to CCDS. Therefore, we believe that the IMR assay to detect plasma NfL levels can only be used as a preliminary screening for CCDS, and further evaluation of the dog’s abnormal behavior to confirm CCDS is also necessary. The measured levels of plasma NfL with IMR in this work differs from that reported by Yun et al. using SIMOA [22]. They observed that the NfL concentration was significantly higher in dogs with meningoencephalitis (serum, 125 pg/mL; CSF, 14,700 pg/mL) than in healthy dogs (serum, 11.8 pg/mL, *p* < 0.0001; CSF, 1410 pg/mL, *p* = 0.0002). The cut-off values were 41.5 pg/mL (serum) and 4005 pg/mL (CSF) for differentiating between healthy dogs and dogs with meningoencephalitis. An inconsistency in the measured levels of plasma biomarkers between using IMR and SIMOA has also been found for amyloid β_1-42_ and total tau protein in humans [23,24,25,26]. This inconsistency in the measured plasma NfL levels could be contributed to by several factors, such as plasma preparation, antibodies used in two different assay technologies, control samples, etc. In addition, our results showed that the plasma NfL level of CCD dogs were 9.37 pg/mL for males and 9.92 pg/mL for females, which was significantly higher than the plasma NfL level of the ND group (7.79 pg/mL for males and 8.02 pg/mL for females, *p* < 0.01). These results indicated that the plasma NfL level increases during normal aging, and these findings are similar to those in dogs and humans [27]. As suggested by Fefer et al. (2022) and Perino et al. (2020), the plasma NfL concentration can be used to quantify the cognitive decline in aging pet dogs [28,29]. According to the CCDS rating scale of 50 points, we divided 36 dogs into 15 with CCDS and 21 without CCDS. Furthermore, using IMR assay, we suggested that the plasma NfL level can be used to quantify cognitive decline in dogs with CCDS.

Past research has found that levels of amyloid β_1-42_ and total tau protein in canine serum cannot be used as biomarkers of neurodegeneration or for the monitoring of aging in dogs because their levels show slight variation across ages [30]. More research is needed as data on whether these blood biomarkers can predict cognitive impairment remains conflicting.

Similar to CCDS, chronic kidney disease (CKD) is a common kidney disease in dogs, especially in older dogs. Symmetrical dimethylarginine (SDMA) can accurately reflect the glomerular filtration rate (GFR), which can be used as a novel renal biomarker. Relford et al. [31] used an IDEXX blood test to find that SDMA in the plasma can be used to diagnose early CKD in dogs and cats. They suggested that SDMA is a more sensitive marker of CKD than creatinine. In this study, using the IDEXX blood test, we found that not only the NfL levels in the plasma of the CCD group were significantly higher than the ND groups, but also the SDMA and creatinine levels in the plasma. Moreover, the biomarkers related to kidney disease in plasma measured in the CCD group, such as blood urea nitrogen, total protein, and albumin, were also significantly higher than those in the ND group. Thus, we suggested that most dogs with CCDS would suffer from kidney disease.

Through biochemical analysis, their results showed that the relevant biomarkers of liver injury, such as AST and ALT, in the plasma of dogs were slightly increased, while the levels of sodium and chloride were significantly reduced compared with young dog controls. In our study, using an IDEXX blood test, we found that the AST, ALT, and GMT levels in the plasma of CCD group were significantly higher than the ND groups. Our data may suggest a possible link between peripheral biomarkers of hepatocellular necrosis and central nervous biomarkers such as NfL in dogs with CCDS. However, more studies are needed to determine whether AST, ALT and GMT levels can be used as biomarker candidates for the early diagnosis of CCDS. Our data from this study clearly show that the C-reactive protein of the CCD group was significantly increased compared with the ND group. The C-reactive protein has been reported as a protein of acute systemic inflammation and could be a biomarker of inflammation [32]. In dogs, C-reactive protein is commonly used to evaluate patients with gastrointestinal disorders such as chronic inflammatory bowel disease [33]. It is possible that older dogs with CCDS may trigger more acute systemic inflammation. Furthermore, our data also shows a significant decrease in sodium and chloride levels, below the physiological range, in the CCD group compared to the ND group. Usually, hyponatremia and hypochloremia are associated with kidney disease, dehydration, vomiting, and diarrhea. Our study has reported that dogs with CCDS often have kidney-related diseases and gastrointestinal disorders. Taken together, dogs with CCDS partially recapitulated the relevant symptoms in the early stages of AD. Furthermore, our data are consistent with other studies showing that NfL concentrations in the plasma can indeed serve as biomarkers for neurodegenerative disease screening in veterinary medicine. However, more research is needed to understand the association of renal and hepatic dysfunction in dogs with CCDS. 

Gaetani et al. [34] reviewed studies on neurofilament structure and function and provided a comprehensive overview of NfL as markers of axonal damage in different human neurological diseases, including multiple sclerosis, neurodegenerative dementia, stroke, trauma brain injury, amyotrophic lateral sclerosis, and Parkinson’s disease. Here, we used IMR assay to explore the NfL as a marker of axonal damage in CCDS. Clinically, the detection of CCDS has multiple meanings. For example, the active treatment of dogs in the early stages of CCDS progression can reduce the cost of treatment and maintain the quality of the dog–owner relationship for a more extended period. It will also allow the veterinarian to determine when to start treating CCDS. Overall, the early diagnosis and care of CCDS can improve animal welfare, so developing IMR assays to detect NfL levels is expected to help develop therapeutic strategies for CCDS.

## 4. Materials and Methods

### 4.1. Sampling Population

Subjects were enrolled at Zaixing Animal Hospital (Taipei City, Taiwan) with the approval of the hospitals’ ethics committees. As shown in Table 1, we selected 36 small- and medium-sized dogs, including 17 male dogs and 19 female dogs, with ages of 3–17, weights of 2.1–8.6 kg, and belonging to 8 different breeds. All animal owners agreed to participate in this research and the owners signed their full informed consent. All animal experiments were in accordance with the Council of Agriculture Executive Yuan Guidelines for the Care and Use of Laboratory Animals. The animal experiments were approved by the Animal Care and Use Supervision Committee of National Taiwan Normal University (Permit number: NTNU/Animal Use/No. 111033).

### 4.2. Behavioral Scoring for Cognitive Impairment

To clearly distinguish dogs between physiological and pathological aging, we used the CCDR developed by Salvin et al. [5] in Table 4. The CCDS rating scale is an assessment tool for the clinical and behavioral screening of CCDS. Combined with veterinary evaluation, it is possible to distinguish between those neurobehavioral changes associated with cognitive impairment in dogs and the behavioral changes of normal aging. The CCDS rating scale primarily assesses disorientation (such as dazedness and losing their way home), memory loss (such as lack of knowledge of the owner and soiling the house), apathy (such as reduced contact time with the owner), and impaired sense of smell (such as difficulty in finding food). These behavioral changes resemble the demented behaviors that have been seen in humans with AD and impair the dog’s quality of life and relationship with its owner. 

### 4.3. Screening NfL Using IMR Assay

IMR is a method that uses recognizable NfL magnetic nanoparticles to analyze the concentration of tiny amounts of NfL in a sample. Under external multiple alternating current (ac) magnetic fields, magnetic nanoparticles oscillate with the multiple ac magnetic fields via magnetic interaction. Thus, the reagent under external multiple ac magnetic fields shows a magnetic property called mixed-frequency ac magnetic susceptibility Xac. Via the antibody on the outermost shell, magnetic nanoparticles associate with and magnetically label bio-molecules (e.g., antigens) to be detected. Due to this association, magnetic nanoparticles become either larger or clustered. Thus, the Xac of the reagent is reduced due to the association between magnetic nanoparticles and detected bio-molecules. This is why the method is referred to as IMR. 

To achieve specific binding between magnetic nanoparticles and NfL, as suggested in our previous study [35], we covalently bound an antibody against NfL (sc20011, Santa Cruz, Dallas, TX, USA) to dextran, which served as an interface between the antibody and the Fe_3_O_4_ core of the nanoparticles. MF-NFL-DOGA is made by covalently binding an anti-NfL antibody to dextran-coated magnetic particles (MF-DEX-0060, MagQu, New Taipei, Taiwan). Briefly, NaIO4 solution is used to oxidize dextran to produce an aldehyde group (-CHO). The dextran can react with the antibodies via the -CH = N-linkage. The unbound antibodies are separated from the solution using magnetic separation. Antibody-functionalized magnetic nanoparticles are well dispersed in PBS solution (pH = 7.4). The average diameter of the magnetic nanoparticles was 53 nm as measured by laser dynamic scattering, and the NfL reagent was stored at 4 °C. A mixture of 60 µL of NfL reagent and 60 µL of sample for IMR measurements was used. An AC susceptibility instrument (XacPro-S, MagQu, New Taipei City, Taiwan) based on a superconducting quantum interference device was used to detect the IMR signal of the sample [35]. 

Before screening NfL levels in canine blood using IMR assay, we collected 5 mL of venous blood from elderly and young control dogs and placed them in serum separation tubes. The blood was centrifuged at 1500–2500 gw for 15 min at room temperature. A total of 500 µL of plasma was aliquoted into each 0.5 mL cryo tube and store at −20 °C or −80 °C. Plasma is required to be frozen no later than 3 h after the blood draw. Each frozen plasma aliquot was placed on wet ice and IMR measurements were completed before it returned to room temperature. A mixture of 60 μL of NfL reagent and 60 μL of blood sample were used for the IMR measurement. A superconducting quantum interference device-based alternative-current (ac) magnetic susceptometer (XacPro-S, MagQu) was used to detect the IMR signal of the blood sample. During the detection of the IMR signal of a sample, two control solutions were used. One was blank, i.e., PBS solution, and the other contained 50 pg/mL NfL (Ab224840, Abcam, Cambridge, UK) in PBS solution. Continuous variables for each measurement are presented as mean ± standard deviation.

We spiked phosphate-buffered saline (PBS) solutions with different concentrations of NfL (Ab224840, Abcam) for studying the concentration-dependent IMR signal of NfL. The limit of blank and the limit of detection were investigated according to the guidelines for evaluating the detection capacity of clinical laboratory measurement procedures. The limit of blank used for assaying NfL with the IMR NfL reagent was 0.001 fg/mL. The limit of detection for assaying NfL using IMR was 0.18 fg/mL according to the following equation: The limit of detection = The limit of blank + 1.65σS, where σS denotes the standard deviation of the measured NfL concentrations of the NfL solutions spiked with a fixed NfL concentration (e.g., 1.0 fg/mL in the current case) in PBS. The concentration range of spiked NfL samples was from 0.001 to 10,000 pg/mL and the measured IMR signal expressed as IMR (%) increased from 1.967 for 0.001 pg/mL NfL to 4.487 for 10,000 pg/mL NfL. 

### 4.4. Clinical Observations and Blood Tests

All dogs underwent clinical and neurological examination (including the assessment of mental status, posture and gait, cranial nerves, postural responses, spinal reflexes, pain, and spinal cord palpation). Experienced veterinarians collected 5 mL of venous blood from elderly and young control dogs and placed them in serum separation tubes. The blood was left to stand at room temperature for 15–30 min, then centrifuged at 3500 × for 10 min. Separated serum was collected, divided into 1.0 mL aliquots, and stored at −80 °C. All serum samples were tested for blood chemistry immediately after sampling, and the remaining serum samples were frozen at −80 °C for later use. A total of 34 individual dog serum samples were tested in this study. All samples were submitted to IDEXX (Westbrook, ME, USA) commercial reference laboratories for analysis with the consent of the pet owners. 

All complete blood counts [the selected parameters were hematocrit, hemoglobin, white blood cell count, eosinophil count, granules count, and platelet count] were performed at IDEXX reference laboratories using the most advanced hematology technology available. Fluorescent laser flow cytometry methods and veterinary specific algorithms make it possible to provide automated complete blood count results with extremely high accuracy. The IDEXX complete blood counts are the appropriate and most cost-effective option for routine preanesthetic or preventive care screenings on clinically healthy animals. Available for canine, feline, and equine patients, this option provides automated complete blood counts, including hemogram, five-part differential, platelets, and platelet indices, as well as (for canine and feline patients) reticulocyte count and reticulocyte hemoglobin.

Plasma biochemical analysis [the selected biochemical parameters were glucose concentration, symmetric dimethylarginine (SDMA), creatinine, urea nitrogen, total protein, albumins, globulins, aspartate transaminase (AST), alanine transaminase (ALT), alkaline phosphatase (ALP), glutamyl transferase (GMT), bilirubin, cholesterol, C-reactive protein, phosphorus (P), calcium (Ca), sodium (Na), potassium (K), and chloride(Cl)] performed at IDEXX reference laboratories used the Catalyst One analyzer of the IDEXX VetLab suite of analyzers, all of which were wired to IDEXX VetLab Station, IDEXX’s laboratory information management system. Connecting several analyzers to the IDEXX VetLab Station can provide you with comprehensive animal patient health information, view the detection results of multiple analyzers from a single report, determine disease progression through parameter trend analysis, and more functions. An elevated SDMA concentration is a reflection of an impaired glomerular filtration rate. Both primary kidney disease and secondary kidney insults, such as concurrent disease, can cause an elevation in SDMA concentration. This algorithm can be followed to investigate elevated SDMA concentrations and determine whether acute, active, or chronic injury occurs. The algorithm can also be used to follow up on an increased SDMA with further investigation, management, and monitoring with a protocol.

### 4.5. Statistical Analysis

The normality of the NfL concentration data was checked using the Kolmogorov–Smirnov test, which revealed a violation of the assumption of normality (*p* = 0.033). The Mann–Whitney U test was used to compare the NfL concentrations between the two groups (i.e., ND and CCD groups). To clarify the difference between the ND and CCD groups, receiver operating characteristic curve analysis was performed. Data obtained using the CCDS rating scale and IMR assay were presented as mean ± standard error of the mean (SEM). Differences between the ND and CCD groups were assessed using one-way or two-way analysis of variance (ANOVA) followed by a Student–Newman–Keuls multiple comparisons posttest. If significant F-values were obtained, significance was defined as *p* < 0.05.

## 5. Conclusions

Using IMR assay, our results showed that the plasma NfL levels of dogs with CCDS significantly increased compared to those without CCDS. Using plasma biochemical analysis, we further confirmed that liver and renal dysfunction biomarkers of dogs with CCDS were significantly elevated compared to normal dogs. CCDS is the canine analog of human AD. Thus, we suggested that plasma NfL should be an ideal blood-based biomarker for the early diagnosis of CCDS.

## Figures and Tables

**Figure 1 ijms-24-13771-f001:**
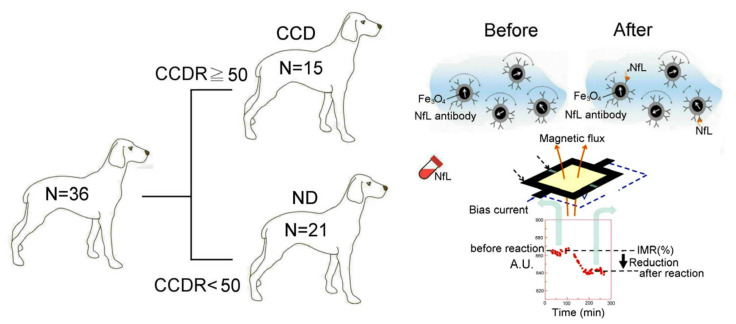
Dog grouping and experimental preparation. According to CCDS rating scale of 50 points, we divided 36 dogs into 15 with CCDS (CCD) and 21 without this disease (ND). Then, a 36-channel high-Tc SQUID alternating current (ac) magnetosusceptometer was used to perform plasma NfL measurement.

**Figure 2 ijms-24-13771-f002:**
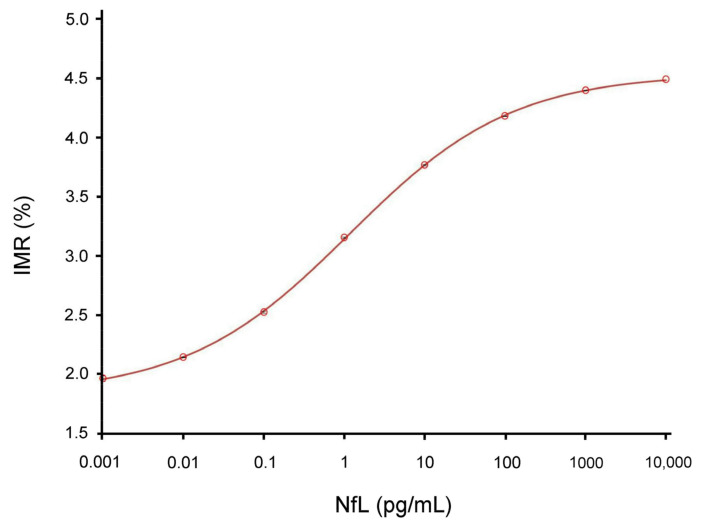
Relationship curve between the IMR signal, IMR (%), and the spiked NfL concentrations. Using a highly sensitive 36-channel high-Tc SQUID alternating current (ac) magnetosusceptometer, the curve of the relationship between IMR signals and plasma NfL expression was established, indicating that plasma NfL expression is positively correlated with IMR signals (as percentages) at concentrations between 0.1 and 10,000 pg/mL, following the logistic function.

**Table 1 ijms-24-13771-t001:** Comparison of characteristics and plasma NfL levels between ND and CCD groups.

	ND (N = 21)		CCD (N = 15)	
Sex (number)	M: 10 (48%)	F: 11 (52%)	M: 7 (47%)	F: 8 (53%)
Age (years) (Range)	7.32 ± 3.16 (3–16)	7.68 ± 3.64 (3–14)	14.37 ± 3.03 ** (12–17)	13.87 ± 3.22 ** (10–17)
Body weight (kg) (Range)	4.2 ± 2.2 (2.1–7.2)	4.5 ± 1.2 (2.3–7.5)	5.2 ± 2.7 (3.3–8.6)	4.9 ± 2.5 (2.8–8.4)
NfL levels (pg/mL) (Range)	7.97 ± 1.36 (5.77–10.37)	8.02 ± 1.63 (6.14–10.67)	9.37 ± 1.70 ** (7.63–13.03)	9.92 ± 1.05 ** (8.03–11.40)

Data are shown as mean ± SEM (** *p* < 0.01, two-way ANOVA followed by Student–Newman–Keuls multiple comparisons posttest).

**Table 2 ijms-24-13771-t002:** Comparison of canine complete blood count and serum biochemical parameters between ND and CCD groups.

Biochemical Parameters	ND (N = 21)	CCD (N = 15)
Hematocrit (%)	42.49 ± 7.23	39.83 ± 7.67
Hemoglobin (g/dL)	14.66 ± 2.64	13.60 ± 2.48
WBC (K/µL)	10.11 ± 2.19	10.18 ± 2.52
White blood cell count (K/µL)	6.36 ± 3.22	7.27 ± 3.74
Eosinophil count (K/µL)	1.21 ± 0.85	1.22 ± 0.73
Granules count (K/µL)	7.64 ± 2.09	7.38 ± 2.12
Platelet count (K/µL)	310.53 ± 83.61	313.58 ± 87.66
Glucose concentration (mg/dL)	94.68 ± 13.19	96.69 ± 13.13
Symmetric dimethylarginine (µg/dL)	9.24 ± 3.12	13.17 ± 3.23 *
Creatinine (mg/dL)	1.27 ± 0.32	1.98 ± 0.28 *
Urea nitrogen (mg/dL)	23.13 ± 8.16 *	29.75 ± 831 *
Total protein (g/dL)	60.27 ± 4.26	69.89 ± 4.22 *
Albumins (g/dL)	32.86 ± 3.32	41.78 ± 3.95 *
Globulins (g/dL)	3.83 ± 1.15	3.93 ± 1.21
Aspartate transaminase (AST/GOT) (U/L)	40.26 ± 3.72	48.57 ± 3.52 *
Alanine transaminase (ALT/GPT) (U/L)	53.89 ± 7.11	108.21 ± 6.79 **
Alkaline Phosphatase (ALP) (U/L)	86.45 ± 7.98	106.50 ± 9.12 **
Glutamyl transferase (GMT) (U/L)	9.53 ± 0.66	16.67 ± 0.72 **
Bilirubin (mg/dL)	0.21 ± 0.15	0.23 ± 0.15
Cholesterol (mg/dL)	198.22 ± 25.2	237.42 ± 26.3 *
C-reactive protein (CRP) (mg/dL)	0.65 ± 0.17	2.39 ± 0.17 **
Phosphorus (mg/dL)	3.90 ± 1.09	3.74 ± 1.04
Calcium (mg/dL)	9.76 ± 0.45	9.95 ± 0.47
Sodium concentration (mmol/L)	138.00 ± 2.29	113.58 ± 2.45 *
Potassium concentration (mmol/L)	4.51 ± 1.78	4.44 ± 1.69
Chloride concentration (mmol/L)	115.33 ± 6.42	93.25 ± 6.67 *

Data are shown as mean ± SEM (** *p* < 0.01, * *p* < 0.05, two-way ANOVA followed by Student–Newman–Keuls multiple comparisons posttest).

**Table 3 ijms-24-13771-t003:** The detection limits of NfL concentration in various assay platforms, such as ELISA, SIMOA, ECL, and IMR.

Assay Platform	Limits of Detection	Refs.
ELISA	5–250 pg/mL	[16,17,18]
SIMOA	0.97–2.2 pg/mL	[19,20]
ECL	15.6 pg/ml	[21]
IMR	0.18 fg/mL	[15]

**Table 4 ijms-24-13771-t004:** Canine cognitive dysfunction rating (CCDR) scale with example data for a dog over the threshold (≥50) for query diagnosis.

DOMAINS/ITEMS
A. SPATIAL ORIENTATION [SCORE (0–25)]
1. disorientation in a familiar environment (inside/outside)
2. recognition familiar people and animals inside or outside the house/apartment
3. abnormally respond to familiar object (a chair, a wastebasket)
4. aimlessly wandering (motorically restless during day)
5. reduced ability to do previously learned tasks
B. SOCIAL INTERACTION [SCORE (0–25)]
1. changes in interaction a man/dog, dog/other dog (playing, petting, welcoming)
2. changes in individual behavior of dog (exploration behavior, play, performance)
3. response to commands and ability to learn new tasks
4. irritability
5. expression of aggression
C. SLEEP-WAKE CYCLES [SCOREx2 (0–20)]
1. abnormally responds in night (wandering, vocalization, motirically restless)
2. switch over from insomnia to hypersomnia
D. HOUSE SOILING [SCORE (0–25)]
1. eliminate at home at random locations
2. eliminate in its kennel or sleeping area
3. changes in signalization for elimination activity
4. eliminate indoors after a recent walk outside
5. eliminate at uncommon locations (grass, concentrate)
TOTAL SCORE (A+B+C+D) (0–95)

a. CLINICAL STAGES: Normal (Score 0–49), Cognitive dysfunction (50–95). b. Frequency: 0—abnormal behavior of the dog was never observed; 2—abnormal behavior of the dog was detected at least once in the last 6 months; 3—abnormal behavior of the dog appeared at least once per month; 4—abnormal behavior of the dog was seen 2–4 times per month; 5—abnormal behavior of the dog was observed several times a week.

## Data Availability

Not applicable.

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
