# Peer review of "Plasma Neurofilament Light Chains as Blood-Based Biomarkers for Early Diagnosis of Canine Cognitive Dysfunction Syndrome"

_ijms, 2023, doi:10.3390/ijms241813771_

Round 1

Reviewer 1 Report

Dear authors,

The authors should better frame the usefulness of this marker and in what context it should and can be used, since it used in isolation does not allow the diagnosis of the disease, given that healthy animals also show it.

The relationship between kidney and liver disease markers does not seem to me to be sufficiently explained or opportune.

I believe that the authors should redefine the age groups with which they worked.

1. Introduction

Line - "seen in older dogs over the age of 10 years that usually show slow behavioral and" Given this, why did you choose "> 9 years" to define the age rank of your sick animals?

2. Results

Lines 77,  87, , 92, and 115 - 42 dogs ????

Lines 78 - 12 normal young dogs (NYD) Normal or healthy young dogs?

Lines 134, 135, and Line 200 - "liver dysfunction biomarkers (such as aspartate transaminase (AST), alanine transaminase (ALT), alkaline phosphatase (ALP), Glutamyl transferase (GMT)),"

Strictly speaking, the first two enzymes are indicators of hepatocellular necrosis and the last two of cholestasis, liver dysfunction is assessed by the bile acid test.

Line 141 - Table 1 . What are the reference values used?

Line 173 - It would be important to study the regularity of the increase in values with age.

4. Materials and Method

Line 233 - 42 or 34 dogs? but in Line 353 - 36????

Line 260 - alternating current (ac)

Line 308 - The caption of figure 5 is not complete in view of the information conveyed therein.

Line 309 -  or SQUID ac magnetosusceptometer ?

Line 326 - veterinary specific

Line 345 - Why does it single out only situations of kidney disease evidenced by an increase in SDMA?

Line 354 - Are dogs under 9 years young? or are they middle aged? It would be better to redefine the age scales. See the age groups of your number 11 reference.

Conclusions

Line 371 -  "for early diagnosis of canine Alzheimer’s Disease." When you talk about early diagnosis, what are you referring to? From what age is screening recommended? What reflection do you anticipate on the sensitivity and specificity of the biomarker under study?

Suggested References:

Use of Cognitive Testing, Questionnaires, and Plasma Biomarkers to Quantify Cognitive Impairment in an Aging Pet Dog Population.Fefer G, Panek WK, Khan MZ, Singer M, Westermeyer HD, Mowat FM, Murdoch DM, Case B, Olby NJ, Gruen ME. J Alzheimers Dis. 2022;87(3):1367-1378. doi: 10.3233/JAD-215562.

Neurofilament light plasma concentration positively associates with age and negatively associates with weight and height in the dog. Perino J, Patterson M, Momen M, Borisova M, Heslegrave A, Zetterberg H, Gruel J, Binversie E, Baker L, Svaren J, Sample SJ. Neurosci Lett. 2021 Jan 23;744:135593. doi: 10.1016/j.neulet.2020.135593. Epub 2020 Dec 24.

Minor editing of English language required

Author Response

Response to Reviewer 1 Comments

Dear Reviewer 1,

Many thanks for valuable comments. In accordance with the concerns you have identified, point-for-point responses to your comments and questions are given below by using red description in our review reports. Also, we have corrected incorrect use in the text and let our manuscript been checked by a English Editing in English writing. Please check them in the revised manuscript. If there is any problem, please address all correspondence concerning the manuscript to Dr. Chung-Hsin Wu.

Reviewer 1 comments

Point 1: The authors should better frame the usefulness of this marker and in what context it should and can be used, since it used in isolation does not allow the diagnosis of the disease, given that healthy animals also show it. The relationship between kidney and liver disease markers does not seem to me to be sufficiently explained or opportune. I believe that the authors should redefine the age groups with which they worked.

Response 1: Many thanks the Reviewer's comments. We have redefined the groups with canine cognitive dysfunction rating (CCDR) in Table 1. Please check them in the revised manuscript.

Point 2: 1. Introduction

Line - "seen in older dogs over the age of 10 years that usually show slow behavioral and" Given this, why did you choose "> 9 years" to define the age rank of your sick animals?

Response 2: Many thanks the Reviewer's comments. We have redefined the groups with CCDR, but not with age rank. Please check them in the revised manuscript.

Point 3: 2. Results

Lines 77,  87, , 92, and 115 - 42 dogs ????

Response 3: Many thanks the Reviewer's comments. We have uniformly corrected the number of dogs to be 36. Please check them in the revised manuscript.

Point 4: Lines 78 - 12 normal young dogs (NYD) Normal or healthy young dogs?

Response 4: Many thanks the Reviewer's comments. We have redefined the groups with CCDR and divided 36 dogs into 21 normal dogs (ND) and 15 old dogs with CCDS (CCD). Please check them in the revised manuscript.

Point 5: Lines 134, 135, and Line 200 - "liver dysfunction biomarkers (such as aspartate transaminase (AST), alanine transaminase (ALT), alkaline phosphatase (ALP), Glutamyl transferase (GMT))," Strictly speaking, the first two enzymes are indicators of hepatocellular necrosis and the last two of cholestasis, liver dysfunction is assessed by the bile acid test.

Response 5: Many thanks the Reviewer's comments. We have corrected "liver dysfunction biomarkers" to " hepatocellular necrosis " in the text. Please check them in the revised manuscript.

Point 6: Line 141 - Table 1 . What are the reference values used?

Response 6: Many thanks the Reviewer's comments. We used IDEXX kits to examine canine complete blood count and serum biochemical parameters. The reference values were used specifically for dogs.

Point 7: Line 173 - It would be important to study the regularity of the increase in values with age.

Response 7: Many thanks the Reviewer's comments. We have studied the regularity of the increase in plasma NfL values with age in Table 1 and Disscusion. Please check them in the revised manuscript.

Point 8: 4. Materials and Method

Line 233 - 42 or 34 dogs? but in Line 353 - 36????

Response 8: Many thanks the Reviewer's comments. We have uniformly corrected the number of dogs to be 36. Please check them in the revised manuscript.

Point 9: Line 260 - alternating current (ac)

Response 9: Many thanks the Reviewer's comments. We have corrected "ac" to " alternating current (ac) " in the text. Please check them in the revised manuscript.

Point 10: Line 308 - The caption of figure 5 is not complete in view of the information conveyed therein.

Response 10: Many thanks the Reviewer's comments. We have integrated Figure 5 into Figure 1 and completed the information of the caption in Figure 1. Please check them in the revised manuscript.

Point 11: Line 309 -  or SQUID ac magnetosusceptometer ?

Response 11: Many thanks the Reviewer's comments. We have corrected "ac magnetosusceptometer" to "alternating current (ac) magnetosusceptometer " in the the caption of Figure 1. Please check them in the revised manuscript.

Point 12: Line 326 - veterinary specific

Response 12: Many thanks the Reviewer's comments. We have corrected " veterinaryspecific " to " veterinary specific " in the text. Please check them in the revised manuscript.

Point 13: Line 345 - Why does it single out only situations of kidney disease evidenced by an increase in SDMA?

Response 13: Many thanks the Reviewer's comments. In Table 2, we found that both renal index creatinine and symmetric dimethylarginine (SDMA) were increased when renal function was lost. Traditionally, renal index creatinine will not increase until more than 75% of renal function is lost, while SDMA is a more sensitive indicator of renal function, and SDMA can rise as early as 25% of renal function was lost Please check them in the revised manuscript.

Point 14: Line 354 - Are dogs under 9 years young? or are they middle aged? It would be better to redefine the age scales. See the age groups of your number 11 reference.

Response 14: Many thanks the Reviewer's comments. We have redefined the groups with CCDR, but not with age rank. Please check them in the revised manuscript.

Point 15: Conclusions

Line 371 -  "for early diagnosis of canine Alzheimer’s Disease." When you talk about early diagnosis, what are you referring to? From what age is screening recommended? What reflection do you anticipate on the sensitivity and specificity of the biomarker under study?

Suggested References:

Use of Cognitive Testing, Questionnaires, and Plasma Biomarkers to Quantify Cognitive Impairment in an Aging Pet Dog Population. Fefer G, Panek WK, Khan MZ, Singer M, Westermeyer HD, Mowat FM, Murdoch DM, Case B, Olby NJ, Gruen ME. J Alzheimers Dis. 2022;87(3):1367-1378. doi: 10.3233/JAD-215562.

Neurofilament light plasma concentration positively associates with age and negatively associates with weight and height in the dog.

Perino J, Patterson M, Momen M, Borisova M, Heslegrave A, Zetterberg H, Gruel J, Binversie E, Baker L, Svaren J, Sample SJ. Neurosci Lett. 2021 Jan 23;744:135593. doi: 10.1016/j.neulet.2020.135593. Epub 2020 Dec 24.

Response 15: Many thanks the Reviewer's comments. As suggested from Fefer et al. (2022) and Perino et al. (2020), plasma plasma NfL concentration can be used to quantify cognitive decline in aging pet dogs. According to CCDS rating scale of 50 points, we divided 36 dogs into 21 normal dogs (ND) and 15 dogs with CCDS (CCD). Furthermore, by IMR assay, we suggested that plasma NfL level can be used to quantify cognitive decline in dogs with CCDS. We have supplemented the description in the Discussion. Please check them in the revised manuscript.

Reviewer 2 Report

Reviewer comments and suggestions

The authors in this detected plasma neurofilament light chains (NfL) as blood-based biomarkers for early diagnosis of canine Alzheimer’s Disease using immunomagnetic reduction (IMR) technology that NfL antibodies were immobilized on magnetic nanoparticles. According to CCDS rating scale, we divided 34 dogs into 12 normal young dogs (NYD), 10 normal old dogs (NOD), and 12 old dogs with CCDS (CCD). By IMR assay, our results showed that the plasma NfL levels of dogs with CCDS were significantly increased than normal dogs (NfL of CCD: 9.88 ± 0.92 pg/ml vs NfL of NYD: 7.63 ± 0.72 pg/ml, and NfL of NOD: 8.58 ± 0.89 pg/ml, P < 0.01). By plasma biochemical analysis, we further confirmed that liver and renal dysfunction biomarkers of dogs with CCDS were significantly elevated than normal dogs (P < 0.01-0.05). We suggested that plasma NfL should be an ideal blood-based biomarkers for early diagnosis of canine Alzheimer’s Disease.

Overall, the manuscript was poorly written. However, major concerns/comments needed to be explained/modified. 

  1. In lines 42-43 Please explain the points on behavioral changes and others
  2. Line 50 “such as neurofilament light 50 protein (NfL)” is this was already reported by another researcher, please explain with the references
  3. Line 53 “In veterinary testing, few studies are conducted on biomarkers associated with CCDS. Biomarkers that are suitable” Please explain extensively here as these were important studies
  4. Line 64-67 Can you please discuss those results here
  5. Figure 3 The legend should be explained
  6. Line 145 Do the authors report about the techniques its sensitivity or specificities
  7. Line 146-147 The first paragraph should be discussing the novelty of your study better to change these lines
  8. Line 192-193 It's not good representation that the first reference was discussed in the discussion section.
  9. All references need to be modified based on MDPI guidelines.

Author Response

Response to Reviewer 2 Comments

Dear Reviewer 2,

Many thanks for valuable comments. In accordance with the concerns you have identified, point-for-point responses to your comments and questions are given below by using red description in our review reports. Also, we have corrected incorrect use in the text and let our manuscript been checked and revised by a English Editing in English writing. Please check them in the revised manuscript. If there is any problem, please address all correspondence concerning the manuscript to Dr. Chung-Hsin Wu.

Reviewer 2 comments

Point 1: Overall, the manuscript was poorly written. However, major concerns/comments needed to be explained/modified.

Response 1: Many thanks the Reviewer's comments. We have corrected incorrect use in the text and let our manuscript been checked and revised by a English Editing in English writing. Please check them in the revised manuscript.

Point 2: In lines 42-43 Please explain the points on behavioral changes and others

Response 2: Many thanks the Reviewer's comments. We have explained the points on behavioral changes in Table 4. Please check them in the revised manuscript.

Point 3: Line 50 “such as neurofilament light protein (NfL)” is this was already reported by another researcher, please explain with the references

Response 3: Many thanks the Reviewer's comments. We have explained eurofilament light protein (NfL) in References 10-11. Please check them in the revised manuscript.

Point 4: Line 53 “In veterinary testing, few studies are conducted on biomarkers associated with CCDS. Biomarkers that are suitable” Please explain extensively here as these were important studies

Response 4: Many thanks the Reviewer's comments. We have explained extensively here as these were important studies that Biomarkers such as plasma NfL that are suitable for routine clinical testing must be readily detectable. NfL concentration can be easily measured in plasma, thus it is a potential biomarker for veterinary testing in the damage of brain tissue [Ref. 11]. Please check them in the revised manuscript.

Point 5: Line 64-67 Can you please discuss those results here

Response 5: Many thanks the Reviewer's comments. Immobilize the NFL antibody on the magnetic nanoparticles, and then detect the NfL concentration in the plasma of normal controls, Parkinson's disease (PD) and AD patients. It was found that there were significant differences in plasma NfL concentrations among normal controls, PD and AD patients [15]. These published results demonstrated the possible clinical impact of using plasma NfL to diagnose AD or PD. Please check them in the revised manuscript.

Point 6: Figure 3 The legend should be explained

Response 6: Many thanks the Reviewer's comments. The legend has been explained as follows: By a highly sensitive 36-channel high-Tc SQUID alternating current (ac) magnetosusceptometer, a curve of the relationship between IMR signals and plasma NfL expression was established that the relationship between plasma NfL expression is positively correlatedwith IMR signals (in percentages) between 0.1 and 10,000 pg/mL which followed a logistic function. Please check them in the revised manuscript.

Point 7: Line 145 Do the authors report about the techniques its sensitivity or specificities

Response 7: Many thanks the Reviewer's comments. We have report about the sensitivity or specificities of plasma NfL detection techniques in second paragraph of the Discussion. Please check them in the revised manuscript.

Point 8: Line 146-147 The first paragraph should be discussing the novelty of your study better to change these lines

Response 8: Many thanks the Reviewer's comments. We have discussed the novelty of our study in first paragraph of the Discussion. Please check them in the revised manuscript.

Point 9: Line 192-193 It's not good representation that the first reference was discussed in the discussion section.

Response 9: Many thanks the Reviewer's comments. We have deleted the first reference was discussed in the discussion section. Please check them in the revised manuscript.

Point 10: All references need to be modified based on MDPI guidelines.

Response 10: Many thanks the Reviewer's comments. We have modified all references based on MDPI guidelines. Please check them in the revised manuscript.

Reviewer 3 Report

Authors investigated the canine cognitive dysfunction syndrome (CCDS)- a progressive neurodegenerative disorder of older dogs, linked to process of aging.

The authors stated that blood samples could be an appropriate method for screening CCDS biomarkers and thus suitable for routine clinical trials.

2.2. Canine CCDR scores -The number of samples is rather low, so this report seems to be a preliminary study.

Relationship curve presented in Figure 3, - standard deviations and statistical data are not reported accurately. The authors should be more precise about the method; it is not enough to refer to a previous study [34].

In addition, the method the authors propose (Figure 5) should be better supported by other scientific data. However, at present the method cannot be suitable for routine clinical testing of dogs.

There are differences in some parameters related to age and/or CCDS. This is a preliminary study for both the number of biological samples and the single method - canine blood IMR assay - set up by the authors and not clinically validated.

The study could be interesting, but there is limited data for the quality of IJMS.

Some authors of the manuscript seem to be present several times in the references (e.g. Yang SY,).  Superfluous self-citations should be avoided.

good

Author Response

Response to Reviewer 3 Comments

Dear Reviewer 3,

Many thanks for valuable comments. In accordance with the concerns you have identified, point-for-point responses to your comments and questions are given below by using red description in our review reports. Also, we have corrected incorrect use in the text and let our manuscript been checked by a English Editing in English writing. Please check them in the revised manuscript. If there is any problem, please address all correspondence concerning the manuscript to Dr. Chung-Hsin Wu.

Reviewer 3 comments

Point 1: Authors investigated the canine cognitive dysfunction syndrome (CCDS)- a progressive neurodegenerative disorder of older dogs, linked to process of aging.

The authors stated that blood samples could be an appropriate method for screening CCDS biomarkers and thus suitable for routine clinical trials.

Point 1: 2.2. Canine CCDR scores -The number of samples is rather low, so this report seems to be a preliminary study.

Response 1: Many thanks the Reviewer's comments. In this study, we have declare that " From our preliminary study, we suggested that IMR technology should be an ideal biosensor in detecting plasma NfL for early diagnosis of CCDS" in the Abstract. Please check them in the revised manuscript.

Point 2: Relationship curve presented in Figure 3, - standard deviations and statistical data are not reported accurately. The authors should be more precise about the method; it is not enough to refer to a previous study [34].

Response 2: Many thanks the Reviewer's comments. We have supplemented the description in the Materials Methods as follows: During detection of the IMR signal of a sample, two control solutions were used. One was blank, i.e., PBS solution, and the other contained 50 pg/ml NfL (Ab224840, Abcam, Cambridge, UK) in PBS solution. Continuous variables for each measurement are presented as mean ± standard deviation. Please check them in the revised manuscript.

IMR Avg

IMR std

1.70725

0.003182

1.96725

0.10147

2.14425

0.024395

2.5275

0.000707

3.156

0.006364

3.76775

0.00601

4.1815

0.00495

4.39725

0.000354

4.4925

0.000707

Point 3: In addition, the method the authors propose (Figure 5) should be better supported by other scientific data. However, at present the method cannot be suitable for routine clinical testing of dogs.

Response 3: Many thanks the Reviewer's comments. It is very important to develop techniques for detecting CDDS in dogs. Thus, we try to detect a plasma neurofilament light chains (NfL) as a blood-based biomarkers for early diagnosis of canine Alzheimer’s Disease using immunomagnetic reduction (IMR) technology that NfL antibodies were immobilized on magnetic nanoparticles. We hope that IMR technology can be widely used in routine clinical testing of dogs.

Point 3: There are differences in some parameters related to age and/or CCDS. This is a preliminary study for both the number of biological samples and the single method - canine blood IMR assay - set up by the authors and not clinically validated.

Response 3: Many thanks the Reviewer's comments. We agree that There are differences in some parameters related to age and/or CCDS. Thus we have redefined the groups with canine cognitive dysfunction rating (CCDR) in Table 1. We have supplemented the description in the Discussion as follows: As suggested from Fefer et al. (2022) and Perino et al. (2020), plasma plasma NfL concentration can be used to quantify cognitive decline in aging pet dogs [29-30]. According to CCDS rating scale of 50 points, we divided 36 dogs into 21 normal dogs and 15 dogs with CCDS. Furthermore, by IMR assay, we suggested that plasma NfL level can be used to quantify cognitive decline in dogs with CCDS. Please check them in the revised manuscript.

Point 4: The study could be interesting, but there is limited data for the quality of IJMS.

Response 4: Many thanks the Reviewer's comments. We provide clinically realistic data for detecting a plasma NfL as a blood-based biomarkers for early diagnosis of canine Alzheimer’s Disease using IMR technology. We believe that the data should meet the quality of the Blood-Based Biomarkers for Alzheimer’s Disease topic of IJMS.

Point 5: Some authors of the manuscript seem to be present several times in the references (e.g. Yang SY,).  Superfluous self-citations should be avoided.

Response 5: Many thanks the Reviewer's comments. We have removed some of the authors' references and avoided superfluous self-citations. Please check them in the revised manuscript.

Round 2

Reviewer 1 Report

Dear authors,

The reference to Alzheimer's disease seems inappropriate, I believe that the most conventional term in use should be privileged.

Line 2 - Title: "Plasma Neurofilament Light Chains as Blood-Based Biomarkers for Early Diagnosis of Canine Cognitive Dysfunction Syndrome."

Line 19 - we divided 36 dogs into 15 dogs with CCDS and 21 dogs without this disease.

Line 41 instead 74 - the CCDS rating (CCDR) scale

Line 95 and 96 - NfL

Line 125 - (ALT), cholestasis (alkaline phosphatase (ALP), Glutamyl transferase (GMT)),

Line 160  - "The cut‐off values were 41.5 pg/mL (serum) and 4005 pg/mL (CSF) for differentiating between healthy dogs and dogs with meningoencephalitis."

Similarly to what was done for meningoencephalitis, the cut-off for CCD should be calculated by you!

The studied marker does not appear only in sick animals, but its value increases in these animals. From what value can we diagnose the disease? In my opinion, this reflection is central to the acceptance of this article. At least the authors should explain the limitations of this work and address what future steps should be taken in this direction.

Line 197 - of liver function !!!!! Please modify according to the latest revision (Point 5)

Line 217 - in different human neurological diseases,

Line 251 - irritability instead of irritable

Point 6 of the previous review was, in our opinion, not correctly answered.

Point 6: Line 141 - Table 1 . What are the reference values used?

Response 6Many thanks the Reviewer's comments. We used IDEXX kits to examine canine complete blood count and serum biochemical parameters. The reference values were used specifically for dogs.

Minor editing of English language required

Author Response

Response to Reviewer 1 Comments

Dear Reviewer 1,

Many thanks for valuable comments. In accordance with the concerns you have identified, point-for-point responses to your comments and questions are given below by using red description in our review reports. Also, we have corrected incorrect use in the text and let our manuscript been checked by a English Editing in English writing. Please check them in the revised manuscript. If there is any problem, please address all correspondence concerning the manuscript to Dr. Chung-Hsin Wu.

Reviewer 1 comments

Point 1: The reference to Alzheimer's disease seems inappropriate, I believe that the most conventional term in use should be privileged.

Line 2 - Title: "Plasma Neurofilament Light Chains as Blood-Based Biomarkers for Early Diagnosis of Canine Cognitive Dysfunction Syndrome."

Response 1: Many thanks the Reviewer's comments. We have changed the title of the manuscript to "Plasma Neurofilament Light Chains as Blood-Based Biomarkers for Early Diagnosis of Canine Cognitive Dysfunction Syndrome." Other similar situations were also corrected. Please check them in the revised manuscript.

Point 2: Line 19 - we divided 36 dogs into 15 dogs with CCDS and 21 dogs without this disease.

Response 2: Many thanks the Reviewer's comments. We have changed the text of the manuscript to "we divided 36 dogs into 15 dogs with CCDS and 21 dogs without this disease." Other similar situations were also corrected. Please check them in the revised manuscript.

Point 3: Line 41 instead 74 - the CCDS rating (CCDR) scale

Response 3: Many thanks the Reviewer's comments. We have changed the text of the manuscript to " the CCDS rating (CCDR) scale." Other similar situations were also corrected. Please check them in the revised manuscript.

Point 4: Line 95 and 96 - NfL

Response 4: Many thanks the Reviewer's comments. NFL in Lines 95 and 96 have been changed to NfL. Other similar situations were also corrected. Please check them in the revised manuscript.

Point 5: Line 125 - (ALT), cholestasis (alkaline phosphatase (ALP), Glutamyl transferase (GMT)),

Response 5: Many thanks the Reviewer's comments. "Cholestasis" has been added in the text. Please check them in the revised manuscript.

Point 7: Line 160  - "The cut‐off values were 41.5 pg/mL (serum) and 4005 pg/mL (CSF) for differentiating between healthy dogs and dogs with meningoencephalitis." Similarly to what was done for meningoencephalitis, the cut-off for CCD should be calculated by you!

Response 7: Many thanks the Reviewer's comments. By IMR assay, our results showed that the plasma NfL levels of dogs with CCDS were significantly increased than those dogs without CCDS (8.01 vs. 9.66 pg/mL, P<0.01).The cut‐off values of plasma NfL levels were 8.84 pg/mL for differentiating between dogs with and without CCDS. We have supplemented the description in the Discussion. Please check them in the revised manuscript.

Point 8: The studied marker does not appear only in sick animals, but its value increases in these animals. From what value can we diagnose the disease? In my opinion, this reflection is central to the acceptance of this article. At least the authors should explain the limitations of this work and address what future steps should be taken in this direction.

Response 8: Many thanks the Reviewer's comments. As suggested from our study, those dogs with NfL levels ≧ 8.84 pg/mL should be at risk for CCDS. However, our experimental data also showed that some dogs with NfL levels ≧ 8.84 pg/mL did not have symptoms related to CCDS, and some dogs with NfL levels < 8.84 pg/mL did have symptoms related to CCDS. Therefore, we believe that the use of IMR assay to detect plasma NfL levels can only be used as a preliminary screening for CCDS, and further evaluation of abnormal behavior of the dog to confirm CCDS is also necessary.

Point 9: Line 197 - of liver function !!!!! Please modify according to the latest revision (Point 5)

Response 9: Many thanks the Reviewer's comments. We have corrected "liver dysfunction biomarkers" to " hepatocellular necrosis " in the text. Please check them in the revised manuscript.

Point 10: Line 217 - in different human neurological diseases,

Response 10: Many thanks the Reviewer's comments. The "human" have been added in the text. Please check them in the revised manuscript.

Point 11: Line 251 - irritability instead of irritable

Response 11: Many thanks the Reviewer's comments. Irritable have been changed to irritability. Please check them in the revised manuscript.

Point 12: Point 6 of the previous review was, in our opinion, not correctly answered. Point 6: Line 141 - Table 1 . What are the reference values used?

Response 12: Many thanks the Reviewer's comments. The reference values used in Table for the IDEXX ProCyte Dx* Hematology Analyzer (as of IDEXX VetLab* Station software version 5.01) have been listed as follow:

Reviewer 3 Report

The revised version of manuscript has been improved, and authors take in account the reviewers comments.

fine

Author Response

Dear Reviewer 3,

Many thanks to the reviewer 3 for valuable comments and recognition. Also, we have let our manuscript been checked by a English Editing in English writing.